# Real-time parameter evaluation of high-speed microfluidic droplets using continuous spike streams

## ABSTRACT

Droplet-based microfluidic devices, with their high throughput and low power consumption, have found wide-ranging applications in the life sciences, such as drug discovery and cancer detection. However, the lack of real-time methods for accurately estimating droplet generation parameters has resulted in droplet microfluidic systems remaining largely offline-controlled, making it challenging to achieve efficient feedback in droplet generation. To meet the real-time requirements, it's imperative to minimize the data throughput of the collection system while employing parameter estimation algorithms that are both resource-efficient and highly effective. Spike camera, as an innovative form of neuromorphic camera, facilitates high temporal resolution scene capture with comparatively low data throughput. In this paper, we propose a real-time evaluation method for high-speed droplet parameters based on spike-based microfluidic flow-focusing, named RTDE, that integrates spike camera into the droplet collection system to efficiently capture information using spike stream. To process the spike stream effectively, we develop a spike-based estimation algorithm for real-time droplet generation parameters. To validate the performance of our method, we collected spike-based droplet datasets (SDD), comprising synthetic and real data with varying flow velocities, frequencies, and droplet sizes. Experiments result on these datasets consistently demonstrate that our method achieves parameter estimations that closely match the ground truth values, showcasing high precision. Furthermore, comparative experiments with image-based parameter estimation methods highlight the superior time efficiency of our method, enabling real-time calculation of parameter estimations.

## CCS CONCEPTS

• **Applied computing** → *Imaging*.

## KEYWORDS

droplet generation, spike camera, real-time evaluation

## 1 INTRODUCTION

Over the past few decades, there has been a persistent increase in the demand for miniaturization of liquid handling due to the rising need for higher throughput and sensitivity in various fields, such as biomedicine, chemistry, life sciences and environmental science[1]. These domains cover a variety of applications, including the diagnosis of rare cells[2], early cancer[3], analysis of enzyme function[4], the study of phenotypic and genetic diversity at the single-cell leve[5]. Droplet microfluidics[6], as a commonly used technique for minimizing sample volumes, offers unprecedented throughput (1-10kHz) compared to robotic liquid handling and digital microfluidics[7]. Moreover, it is cost-effective, consumes minimal reagents, and offers enhanced sensitivity[1] due to its large surface-to-volume ratio. Despite these advantages, achieving rapid response and real-time control of droplet microfluidic control systems in high-speed scenarios remains an immensely challenging task[1]. Conventional one-dimensional acquisition methods are often inadequate in capturing comprehensive droplet information, while image-based acquisition methods require a trade-off between data throughput and time-consumption[8]. Therefore, parameter setting for droplet generation control often relies on predictive understanding[9] of multiphase flows or numerical simulations of fluid dynamics[10]. However, these predictions and simulations often involve a degree of uncertainty[9], leading to an iterative design process. Therefore, the development of a real-time parameter evaluation method is crucial for the design, evaluation, and control of droplet microfluidic platforms.

Real-time parameter estimation of microfluidic droplets requires the implementation of a high throughput detection system. These systems may be classified into conventional one-dimensional detection methods[11–13] and subsequent image-based detection methods[14–17]. **Traditional detection methods**. Typically, one-dimensional detection methods employ cellular-level optical probes for 'one-by-one' measurements. Sample detection is typically achieved by optical characteristics such as fluorescent light or chromogenic reaction, and is detected using photomultiplier tubes (PMTs) or photodiodes[18]. Although this approach allows for high-throughput and quantitative droplet detection, it often sacrifices spatial resolution in favour of throughput and lacks the capability to record morphological properties of droplets with varying encapsulated contents[1]. **Image-based detection methods**. Image-based detection techniques can be broadly classified into two primary categories: camera-based methods[16, 19–21] and scanning-based methods[22–25]. Camera-based detection systems typically utilize conventional two-dimensional arrays such as CCD or CMOS arrays for data acquisition. These systems are capable of capturing two-dimensional images given a sufficient exposure time. However, the presence of this exposure time poses challenges when applied to high-speed acquisition in droplet microfluidics[18]. Despite attempts to mitigate these challenges by narrowing the field of view or employing spatiotemporal multiplexing strategies[24, 25], the high-throughput data output often impedes real-time analysis and processing. In contrast, scanning-based methods integrate PMTs with laser spot scanning, followed by post-processing to generate

*ACM MM, 2024, Melbourne, Australia*

© 2024 Copyright held by the owner/author(s). Publication rights licensed to ACM.
ACM ISBN 978-x-xxxx-xxxx-x/YY/MM
https://doi.org/10.1145/nnnnnnn.nnnnnnn

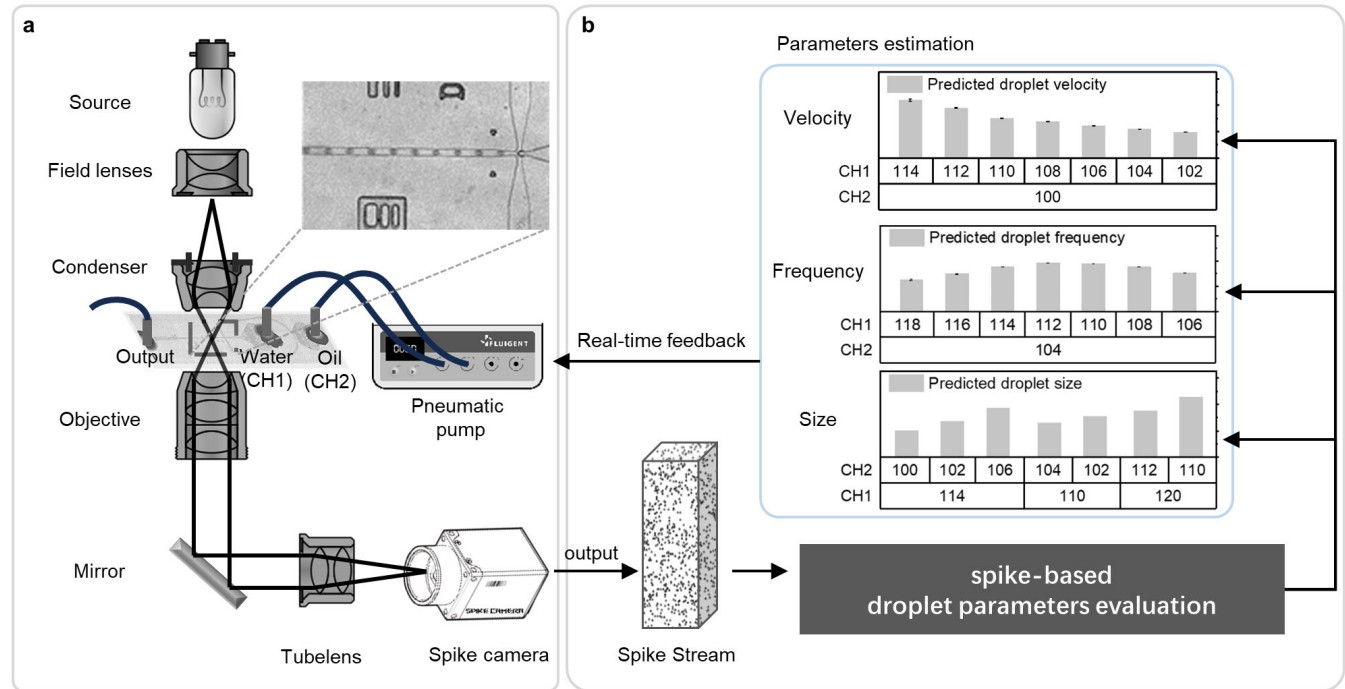

Figure 1: (a) The spike-based microdroplet acquisition system comprises a microscope with a spike camera and a microfluidic droplet device. The microfluidic device has two-phase inputs: oil and water, which are used to generate droplets by shearing water with oil. (b) The spike camera outputs spike streams, which are then processed in real-time using a spike-based droplet parameter estimation algorithm to extract information regarding droplet frequency, velocity, and size. This information is then fed back to fine-tuning the input parameters.

intensity images[8]. In comparison to camera-based methods, scanning methods offer superior bandwidth and lower dark noise. Nevertheless, overall throughput is constrained by scanning speed and achieving real-time analysis and data processing remains a significant challenge[8]. To effectively estimate the real-time parameters of microfluidic droplets, several factors need to be considered: **(1) Sampling rate of detection system**. In the dynamic environment of droplet microfluidics, evolving high-velocity motion, it is imperative that the data acquisition system swiftly captures the scene information. Comprehensive data collection is a prerequisite for effective and reliable subsequent parameter analyses. **(2) Bandwidth of detection system**. Efficient use of bandwidth in data collection facilitates real-time analysis and processing, as opposed to the conventional practice of data storage and subsequent processing. **(3) Real-time parameter estimation algorithm**. Given the collected data, the primary objective is to perform parameter estimation based on the intrinsic characteristics of the data, thus minimizing analysis time and meeting real-time requirements.

In recent years, neuromorphic cameras[26–30] have emerged as a novel type of imaging technology, boasting asynchronous operation at each pixel and outputting data in a discrete format. This unique design enables them to offer remarkable imaging capabilities, particularly excelling in high temporal resolution with relatively small bandwidth. They have shown excellent performance in various high-speed applications. Two common types are event cameras[26, 28, 30] and spike cameras[27, 29]. Event cameras trigger events asynchronously based on set thresholds of light intensity changes, allowing for the capture of dynamic scene information at extremely high temporal resolutions. However, while they excel at capturing dynamic information in scenes, their ability to fully reconstruct scene details remains limited. Although some research endeavours to recover intensity from event streams[31–34], the reconstruction performance still exhibits considerable gaps compared to traditional cameras. While event cameras also hold the potential for high-frequency analysis of droplet parameters, in practical applications, the need to obtain high-quality images at any given moment alongside real-time parameter estimation favours spike cameras over event cameras for this task. Spike cameras operate by asynchronously integrating each pixel, firing a spike when a set threshold is reached. They can record rich spatiotemporal information with extremely high temporal resolution and subsequently recover intensity information at any given moment[29]. Compared to other sensors, spike cameras can capture more comprehensive spatiotemporal information in droplet microfluidic scenes with relatively small bandwidth. This facilitates precise estimation of droplet generation parameters in subsequent stages.

In this paper, we introduce spike camera to parameter estimation in liquid microfluidics (Figure 1(a)) for the first time, demonstrating the potential application of spike camera in high-speed microfluidic scenarios. In contrast to the discrete two-dimensional image output

of traditional cameras, spike cameras generate continuous spike streams containing rich spatiotemporal information. To fully leverage this advantage, we have developed a spike-based method for real-time estimating droplet parameters (RTED) directly from the characteristics of the spike stream (Figure 1(b)) . This innovative approach eliminates the need for image reconstruction from spike streams, resulting in outstanding temporal efficiency. When combined with the efficient data acquisition capabilities of the spike camera, real-time estimation and feedback of droplet parameters become feasible. Our method demonstrates high precision through experiments conducted on both simulated and real-world data. Furthermore, comparative experiments with image-based parameter estimation methods highlight the exceptional temporal efficiency of our method, approaching the stringent requirements of real-time computation. In summary, our contributions can be summarized as follows:

- We are the first to introduce spike cameras to information acquisition in droplet microfluidics. In contrast to existing methods that require trade-offs between bandwidth and acquisition speed, spike camera enables high temporal resolution spatial information acquisition with relatively low bandwidth. This breakthrough holds promise in overcoming the traditional separation of data acquisition and analysis in droplet microfluidics, achieving real-time estimation and feedback seamlessly integrated into a single system. Ultimately, this will achieve real-time, relatively high-precision parameter estimation of droplet parameters with relatively low bandwidth.
- We propose an end-to-end spike-based droplet parameter estimation method, without the need for image reconstruction from spike streams. This method boasts exceptionally high temporal efficiency, capable of meeting real-time processing requirements. Experiments conducted on both synthesized and real-world data consistently demonstrate its superior accuracy.
- We collected and open-source the first spike-based droplet datasets (SDD), containing both synthetic and real-world data with varying flow velocities, frequencies, and droplet sizes.

## 2 RELATED WORK

### 2.1 Droplet Parameter Estimation

A droplet microfluidic system is a system capable of generating droplets at a microscale by manipulating fluids. The precise control of droplets relies not only on the properties of the input fluids but also on the design of microfluidic structures[35]. To meet the practical demands of droplet generation, it's often essential to fine-tune input parameters and optimize the design of microfluidic structures based on estimated parameters[1]. In traditional one-dimensional data acquisition systems, data collection typically only allows for quantitative analysis of the components within droplets, posing challenges in acquiring morphological characteristic parameters of the droplets[15], thus rendering it unsuitable for parameter estimation in droplet microfluidics. Image-based methods offer significant advantages in this scenario. By utilizing captured image data combined with conventional digital image processing methods,

relevant parameters of droplet generation can be extracted from the images[18]. However, to achieve continuous data acquisition in high-speed scenarios, the system requires high data throughput. In such cases, it becomes challenging to promptly utilize the collected data for practical parameter estimation, resulting in a lack of synchronization between data collection and parameter estimation, or requiring offline parameter estimation. This asynchrony severely affects the convenience of droplet microfluidic regulation and makes it difficult to promptly respond to fluctuations during actual operation. In recent years, the integration of deep learning into microfluidic applications has shown promising performance in various aspects such as microfluidic design[9, 36, 37], control[9, 38], and analysis[39–41]. However, deep learning training typically requires a large amount of data, and microfluidic experiments involve expensive equipment and complex operations, making data acquisition still a challenging task. This limitation results in the effectiveness of most models remaining limited to simple or specific scenarios, thus hindering widespread application. Additionally, the training and inference processes of these models are often time-consuming, failing to meet the real-time requirements essential for droplet control in microfluidics[1]. Hence, there is an urgent need for a method capable of realizing real-time estimation and feedback of droplet parameters.

### 2.2 Spike-based Image Reconstruction

Spike cameras enable continuous recording of photons in a scene through asynchronous spike emission, offering exceptionally high temporal resolution and dynamic range. The naive approach to estimating droplet parameters from spike streams involves directly reconstructing them into grayscale images and subsequently applying traditional image-based parameter extraction algorithms. Several studies have endeavoured to reconstruct grayscale images from spike streams, which fall into three main categories: statistic-based methods[42], bio-inspired methods[43, 44], and deep learning methods[45, 46]. Statistic-based methods mainly include texture from playback (TFP) and texture from inter-spike intervals (TFI), with TFI offering higher temporal resolution compared to TFP. Bio-inspired methods primarily enhance reconstruction quality by incorporating biological principles but struggle to balance noise and motion blur simultaneously. Deep learning-based methods can achieve superior reconstruction quality but typically require extensive labelled datasets for training and still face challenges in generalization. Some research[46] has attempted to achieve reconstruction quality comparable to supervised methods using self-supervised methods, which do not require labelled data for training and offer greater convenience. However, these methods still require considerable time consumption, making it difficult to match the rapid response and real-time performance requirements of droplet control. Therefore, the development of efficient droplet parameter estimation methods based on the inherent characteristics of spike streams remains profoundly significant.

## 3 SPIKE GENERATION MECHANISMS

The spike camera converts the photons in space into an electrical current through a photoelectric converter, which is accumulated in capacitors, forming an integral sampling. When the voltage on

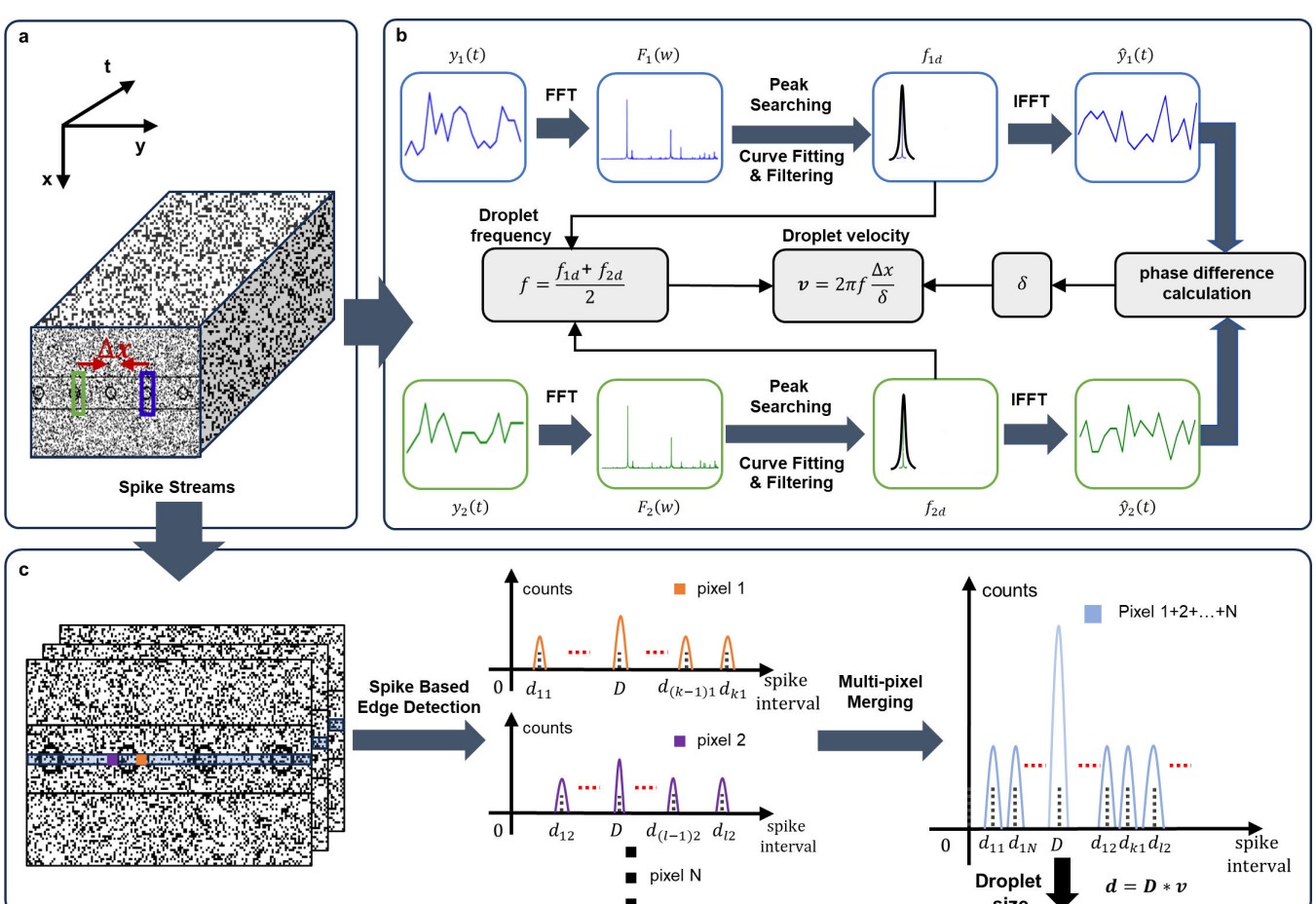

**Figure 2: A schematic diagram of the spike-based droplet parameter estimation algorithm. Based on the inherent characteristics of (a) the spike stream, extracting (b) the frequency, velocity, and (c) size of the droplets. The frequency of the droplets is directly extracted from the frequency domain features of the spike stream, while the velocity of the droplets is calculated based on the correlation of signals to determine the transit time, combined with the known distance $\Delta x$. The size of the droplets is obtained based on statistical features of the spike stream, where $d_{12}$ represents the interval between the first edges of the droplets statistically calculated from the second pixel, and similar for others. $D$ represents the spike interval corresponding to the droplet size.**

the capacitor exceeds a set threshold, a spike is fired, followed by a capacitor reset. This process can be represented as follows:

$$\int_{t}^{t+\Delta t} I(t)dt \geq \vartheta \tag{1}$$

where $\Delta t$ represents the readout period of the spike camera, $I(t)$ denotes the electric current, and $\vartheta$ is the set threshold. When the current period exceeds this threshold, a spike is fired. Typically, a spike sensor consists of an array of $H \times W$ pixels, and during high-speed acquisition periods, the output spike stream forms an $H \times W \times T$ matrix. In contrast to event cameras, which focus solely on capturing dynamic information, spike cameras provide a complete record of scene information. The spike stream contains a wealth of spatiotemporal information, thereby offering a more detailed reference for subsequent processing.

## 4 METHOD

### 4.1 Droplet generation frequency evaluation

As spike streams inherently contain rich frequency information, the droplet generation frequency information can be directly extracted from them, as shown in Figure 2(b). To begin, a rectangular block $B_L$ with $H \times W$ pixels is designated at the location where the droplets traverse the microfluidic channel. The channel position can be identified from the image obtained by direct summation of spike streams. Here, the length of $H$ should be greater than the width of the microfluidic channel, while the size of $W$ should be smaller than the minimum value between the droplet spacing and droplet size. Since the droplet spacing and droplet size are unknown, the $W$ is usually set as 5 pixels. Subsequently, the spike streams within this block along the time domain are extracted, and this could be

expressed as:

$$S = \{s_1(t), s_2(t), \ldots, s_n(t)\} \quad (2)$$

where $n \in [1, 2, \cdots, H \times W]$ and $s_i(t)$ denotes the spike streams at the $i - th$ point within block $B_L$. To mitigate the impact of background noise on the calculation process, the spike within the block $B_L$ is accumulated at each point in time according to the following equation:

$$y(t) = \sum_{i=1}^{H*W} S = \sum_{i=1}^{H*W} s_i(t) \quad (3)$$

Next, the cumulative spike data $y(t)$ is subjected to Fourier transform to obtain the power spectrum within a designed time range $cT$ (where $T$ represents the sampling period of the spike camera). This computation could be formulated as follows:

$$psd(w) = \lim_{c \to \infty} \frac{|F_c(\omega)|^2}{c} = \lim_{c \to \infty} \frac{\left| \sum_{n=0}^{c-1} y(nT) e^{-\frac{j2\pi nw}{c}} \right|^2}{c} \quad (4)$$

where $c$ is typically set to 20000 in this paper, owing to the 50 $us$ for the data readout period of the spike camera. The power spectrum $psd(w)$ primarily comprises the fluctuation frequency $f_n$ of background noise and its higher harmonics. The magnitude of this component is generally determined by the illumination intensity and the threshold for spike firing in the spike camera. Furthermore, the dominant frequency pertains to the droplet generation frequency $f_d$, which functions as a carrier wave overlying the background noise frequency $f_n$. By analyzing the illumination intensity in a background area, the frequency $f_n$ range associated with background noise in the power spectrum could be determined. Peaks are then sought within the background frequency value $f_n$ in the power spectrum. A curve fitting process is performed at the location of the peak, and the fitted curve's peak position corresponds to the droplet generation frequency $f_d$.

## 4.2 Droplet velocity evaluation

Based on the principle of signal correlation, the velocity of droplets in microfluidic channels can be evaluated, as shown in Figure 2(b). This approach entails transforming the droplet velocity into the assessment of phase for spike streams at two distinct positions along the microfluidic channel. Initially, two rectangular blocks $B_{L1}$ and $B_{L2}$ positioned on the microfluidic channel with a spacing of $\Delta x$ should be selected. The channel position can be identified from an image obtained by direct summation of spike streams. These blocks have uniform dimensions $H * W$ pixels, with constraints akin to those outlined for the block $B_L$ in Section 4.1, and the spacing $\Delta x$ should not exceed the length of one droplet cycle, which is typically kept as small as possible, for example, two pixels. Extracting the spike streams from these two blocks could yield two sets of signals, denoted as $y_1(t)$ and $y_2(t)$, expressed as shown in Eq.(3). Subsequently, we calculate the power spectra of $y_1(t)$ and $y_2(t)$. By employing the method described in section 4.1, we can determine the droplet generation frequencies, $f_{d1}$ and $f_{d2}$, corresponding to the signal $y_1(t)$ and $y_2(t)$, which typically remain reasonably consistent.

To mitigate the impact of noise on the computation of the droplet generation signal phase, we further perform the Fourier transform on $y_1(t)$ and $y_2(t)$, which could be calculated by the following

formula:

$$F_1(w) = \sum_{n=0}^{N-1} y_1(nT) e^{-\frac{j2\pi nw}{c}} \quad (5)$$

$$F_2(w) = \sum_{n=0}^{N-1} y_2(nT) e^{-\frac{j2\pi nw}{c}} \quad (6)$$

where $N$ represents the length of the signals. Frequencies other than droplet generation frequencies $f_{d1}$ and $f_{d2}$ are filtered out on the frequency domains $F_1(w)$ and $F_2(w)$ respectively. This results in frequency domain signals $F_1(f_{d1})$ and $F_2(f_{d2})$, which are then inverse Fourier transformed back to the time domain and yield signals $\hat{y}_1(t)$ and $\hat{y}_2(t)$ containing solely the droplet generation frequency.

Subsequently, we calculate the cross-correlation of the signal $\hat{y}_1(t)$ and $\hat{y}_2(t)$. For $\hat{y}_1(t)$ and $\hat{y}_2(t)$, we perform a Fourier series expansion, which could be calculated by the following formula:

$$\hat{y}_1(t) = \sum_{i=0}^{n} \Psi_i sin(2\pi f_i t + \phi_i) \quad (7)$$

$$\hat{y}_2(t) = \sum_{j=0}^{n} \Psi_j sin(2\pi f_j t + \phi_j) \quad (8)$$

where $f_0 = 0$ corresponds to the direct current (DC) component (taking into account that we can filter out the DC components from $\hat{y}_1(t)$ and $\hat{y}_2(t)$ and not consider them later). $\Psi_i$ and $\Psi_j$ correspond to the intensity of frequency components $f_i$ and $f_j$ of $\hat{y}_1(t)$ and $\hat{y}_2(t)$ respectively. These intensities can be obtained by performing a Fourier transform on $\hat{y}_2(t)$ and $\hat{y}_2(t)$. $\phi_i$ and $\phi_j$ correspond to the phases of frequency components $f_i$ and $f_j$ of $\hat{y}_1(t)$ and $\hat{y}_2(t)$ respectively. Further calculate the product of $\hat{y}_1(t)$ and $\hat{y}_2(t)$, which could be expressed as:

$$\hat{y}_1(t) \times \hat{y}_2(t)$$
$$= \sum_{i=1}^{n} \Psi_i sin(2\pi f_i t + \phi_i) \times \sum_{j=1}^{n} \Psi_j sin(2\pi f_j t + \phi_j) \quad (9)$$
$$= \sum_{i=1}^{n} \sum_{j=1}^{n} \Psi_i B_j sin(2\pi f_i t + \phi_i) sin(2\pi f_j t + \phi_j)$$

According to the equation $sin(a)sin(b) = -1/2 * [cos(a+b) - cos(a-b)]$, the Eq(9) can be further rewritten as:

$$\hat{y}_1(t) \times \hat{y}_2(t) = \sum_{i=1}^{n} \sum_{j=1}^{n} \frac{1}{2} \Psi_i \Psi_j [cos(2\pi(f_i - f_j)t + \phi_i - \varphi_j)$$
$$- cos(2\pi(f_i + f_j)t + \phi_i + \varphi_j))] \quad (10)$$

The sequence $\hat{y}_1(t) \times \hat{y}_2(t)$ is then integrated over the period $E = t_m - t_1$, where $t_1$ represents the starting time of the spike stream chosen for analysis, while $t_m$ represents the ending time, which could be expressed as:

$$\sum_{t=t_1}^{t_m} \hat{y}_1(t) \times \hat{y}_2(t)$$
$$= \sum_{t=t_1}^{t_m} \sum_{i=1}^{n} \sum_{j=i}^{n} \frac{1}{2} \Psi_i \Psi_j [cos(2\pi(f_i - f_j)t + \phi_i - \varphi_j)$$
$$- cos(2\pi(f_i + f_j)t + \phi_i + \varphi_j))]$$
$$= \sum_{t=t_1}^{t_m} \sum_{i=1}^{n} \sum_{j=i}^{n} \frac{1}{2} \Psi_i \Psi_j cos(2\pi(f_i - f_j)t + \phi_i - \varphi_j) \quad (11)$$
$$+ \sum_{t=t_1}^{t_m} \sum_{i=1}^{n} \sum_{j\neq i}^{n} \frac{1}{2} \Psi_i \Psi_j cos(2\pi(f_i - f_j)t + \phi_i - \varphi_j)$$
$$- \sum_{t=t_1}^{t_m} \sum_{i=1}^{n} \sum_{j=1}^{n} \frac{1}{2} \Psi_i \Psi_j cos(2\pi(f_i + f_j)t + \phi_i + \varphi_j)$$

when $E$ is sufficiently large, the $\sum_{t=t_1}^{t_m} \sum_{i=1}^{n} \sum_{j\neq i}^{n} \frac{1}{2} \Psi_i \Psi_j cos(2\pi(f_i - f_j)t + \phi_i - \varphi_j)$ and $\sum_{t=t_1}^{t_m} \sum_{i=1}^{n} \sum_{j=1}^{n} \frac{1}{2} \Psi_i \Psi_j cos(2\pi(f_i + f_j)t + \phi_i + \varphi_j)$ will only accumulate the value of the last incomplete period due to the presence of the cosine function. In comparison to the integral value of the

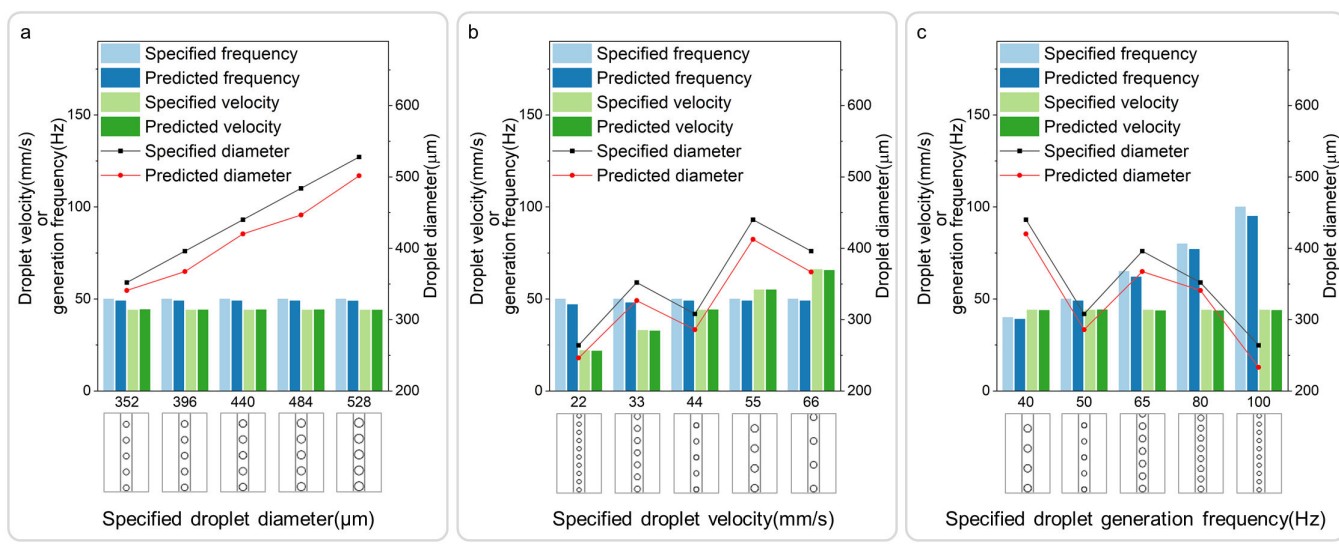

**Figure 3: Results on simulation datasets. (a) The estimation results of our method during the continuous variation of droplet size. (b) The estimation results of our method during the continuous variation of droplet velocity. (c) The estimation results of our method during the continuous variation of droplet frequency**

$\sum_{t=t_1}^{t_m}\sum_{i=1}^{n}\sum_{j=i}^{n}\frac{1}{2}\Psi_i\Psi_j cos(2\pi(f_i-f_j)t+\phi_i-\varphi_j)$, the contribution of these terms could be considered negligible. Therefore, the equation above

$$\sum_{t=t_1}^{t_m}\hat{y}_1(t)\times\hat{y}_2(t)$$
$$\approx\sum_{t=t_1}^{t_m}\sum_{i=1}^{n}\sum_{j=i}^{n}\frac{1}{2}\Psi_i\Psi_j cos(2\pi(f_i-f_j)t+\phi_i-\varphi_j) \quad (12)$$
$$=\sum_{t=t_1}^{t_m}\sum_{i=1}^{n}\frac{1}{2}\Psi_i\Psi_i cos(\phi_i-\varphi_i)=m\sum_{i=1}^{n}\frac{1}{2}\Psi_i\Psi_i cos(\phi_i-\varphi_i)$$

Then, we can obtain the following approximate formula:

$$\sum_{t=t_1}^{t_m}\hat{y}_1(t))\times\hat{y}_2(t))\sim m\sum_{i=1}^{n}\frac{1}{2}\Psi_i\Psi_i cos(\phi_i-\varphi_i) \quad (13)$$

Since only the droplet generation frequency $f_d1$ and $f_d2$ is present in signals $\hat{y}_1(t)$ and $\hat{y}_2(t)$ respectively, leting $\delta=\phi_i-\varphi_i$, we can further obtain:

$$\delta=\pi(f_{d1}+f_{d2})t \quad (14)$$

letting $f=(f_{d1}+f_{d2})/2$, $t$ can be calculated by the following equation:

$$t=\delta/(2\pi f)=\Delta x/v \quad (15)$$

Finally, the velocity of droplet generation can be calculated according to the following equation:

$$v=2\pi f\frac{\Delta x}{\delta} \quad (16)$$

## 4.3 Droplet size evaluation

The size of the droplets can be directly obtained through the statistical characteristics of the spike stream, as shown in Figure 2(c). Two lines of length $L$, parallel to the microfluidic channel are taken inside the channel, but at a certain distance apart. This distance is typically required to be less than half the size of the droplets, which are typically kept as small as possible, for example, two pixels. Then, the spike streams at each pixel on these lines are obtained, which can be expressed as:

$$Y_i=\{y_{1i}(t),y_{2i}(t),...,y_{mi}(t)\} \quad (17)$$

where $i\in\{1,2\}$ represent different lines, $y_{mi}(t)$ denotes the spike stream and $m\in\{0,L\}$. For each pixel's spike stream, we can obtain the spike interval based on the TFI principle, resulting in:

$$D_i=\{d_{1i}(t),d_{2i}(t),...,d_{mi}(t)\} \quad (18)$$

Due to the lower intensity of the droplet's edge compared to the background intensity, differences in spike intervals can arise. Therefore, by setting an appropriate threshold $\Theta$, we can extract the positions where the droplet's edge lies. Taking $d_{1i}(t)$ as an example, we can obtain by:

$$p_{1i}(t)=\mathbb{F}(d_{1i}(t)>\Theta) \quad (19)$$

where $\mathbb{F}(\cdot)$ is a function that sets positions that do not meet the condition to zero and $p_{1i}(t)$ contains values only at positions where the droplet's edge is located. Then, similarly, based on the TFI principle, we can obtain the interval between adjacent edges $q_{1i}(t)$, and further obtain the actual scale that the adjacent edges represent as $l_{1i}(t)$ by:

$$l_{1i}(t)=q_{1i}(t)*v \quad (20)$$

In theory, we can directly derive the size of the droplet from $l_{1i}(t)$. However, in practical scenarios where noise is present, obtaining a continuous edge is often challenging. This can lead to biases in single-pixel statistical values. To mitigate this issue, we can enhance the statistical features by aggregating values from a large number of pixels, Applying the same operation to all pixel points on $Y_i$, we can obtain:

$$L_i=\{l_{1i}(t),l_{2i}(t),...,l_{mi}(t)\} \quad (21)$$

we then conduct statistical analysis on the values in $L_i$. The two values with the highest frequency are respectively considered as the droplet size and the droplet interval. However, directly deducing this from $L_1$ alone is not feasible. Therefore, by comparing the discrepancies in statistical values between $L_1$ and $L_2$, along with their relative positions, we can ascertain the droplet size. Typically, as the distance from the centre of the pipeline increases, the two highest statistical values decrease correspondingly to the size of the droplets, while the increase pertains to the spacing between the droplets.

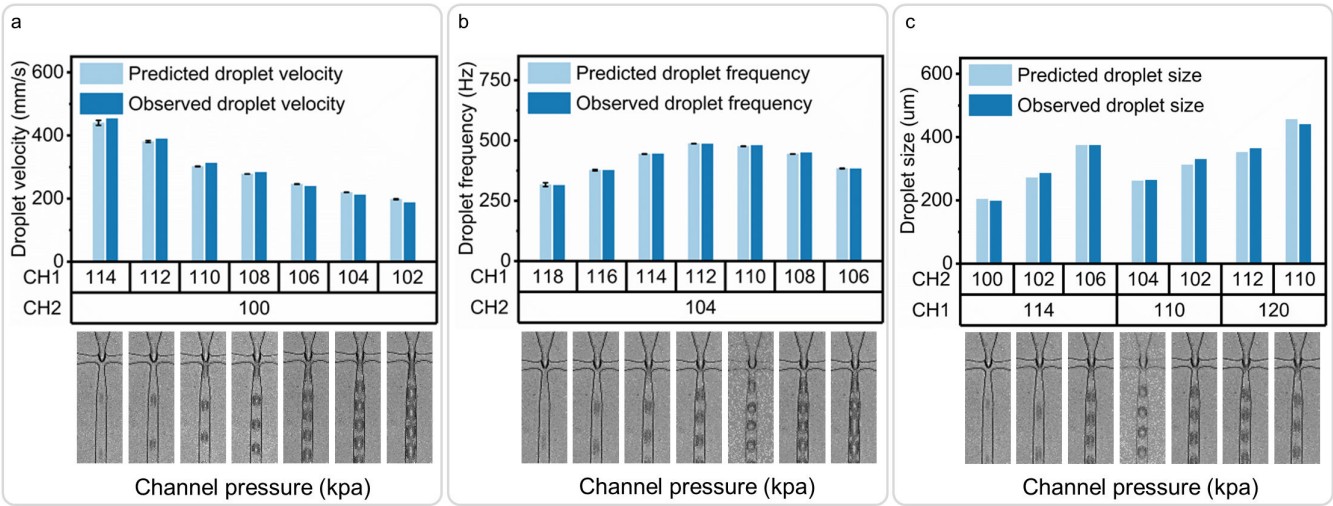

Figure 4: Results on real-world datasets. (a) The comparison results between the droplet velocities estimated by our method and the droplet flow velocities observed in actual measurements when the oil phase was fixed and the water phase parameters were continuously adjusted. (b) The comparison results between the droplet frequency estimated by our method and the droplet frequency observed in actual measurements when the oil phase was fixed and the water phase parameters were continuously adjusted (c) The comparison results between the droplet size estimated by our method and the droplet size observed in actual measurements when the water phase was fixed and the oil phase parameters were continuously adjusted.

## 5 EXPERIMENTS

### 5.1 The Spike-based Droplets Datasets

**Synthetic Dataset:** We simulate droplet morphological features using circles and then simulate the flow process of the droplets by pixel movement to generate a video stream. Subsequently, we assign different sets of parameters including size, flow velocity, and frequency to the droplets, generating multiple sets of data, each containing 20,000 frames. These datasets are then employed as inputs to Spikingsim to simulate spike streams.

**Real-world Dataset:** The real-world data primarily utilized spike cameras mounted on an electric microscope for capturing droplets microfluidics. The microfluidic chip for droplet manipulation has two inputs: oil and water. By shearing water with oil, droplets are formed. Adjusting the input pressure of these two components allows for fine-tuning of droplet generation parameters. In this work, we collected data sets with different input parameters, where the droplets encompass various generation frequencies, flow velocities, and sizes.

### 5.2 Results on Simulation Datasets

To validate the accuracy of our spike-based parameter estimation method, we conducted experiments on simulated data. The primary goal was to confirm whether the parameter values estimated by our method were consistent with the settings employed in the simulation. The simulated dataset consisted of three sets of data, each featuring variations in droplet size, frequency, and velocity. Generally, two parameters were held constant while the third one was adjusted. For example, when adjusting droplet size, we kept the droplet's frequency and velocity constant while simulating various droplet sizes. Similar procedures were followed for the other two sets. As shown in Figure 3, the results of the experiments on simulated data demonstrated that our method's estimated values closely matched the simulated preset values. However, there were slight discrepancies in the estimated values. For instance, when estimating the droplet size, these deviations were

partially attributed to the inherent width of the droplet edge, as our statistical analysis only accounted for the outermost positions. Moreover, the finite time resolution of the camera affected the precision of droplet size estimation. Although our method effectively mitigated the influence of noise, some noise near the droplet generation frequency might persist, influencing the accuracy of the final fitting process and resulting in minor deviations in frequency estimation. Similarly, velocity estimation was influenced by fluctuations in frequency, and slight disparities in droplet frequency between adjacent positions could lead to variations in phase difference calculations. Nonetheless, these deviations were within a small range. Overall, these experiments demonstrated that the results of our parameter estimation method were consistent with theoretical expectations.

### 5.3 Results on Real-world Datasets

To validate the performance of our parameter estimation method on real-world data, we collected droplet data under different input parameters by adjusting the pressure of the oil and water phases. This included data where the oil phase was fixed while the water phase was continuously adjusted, as well as data where the water phase was fixed while the oil phase was continuously adjusted. The comparison between the droplet parameters estimated by our method and the observed droplet parameters under different input parameters is shown in Figure 4. It demonstrated that the parameters predicted by our method are consistent with the observed parameters. However, due to the presence of noise, there is slight degree of fluctuation in the predicted results, but they still remain close to the observed values. Comparatively, the calculation of droplet size is influenced by both velocity and time resolution. This results in the size detection resolution being determined by the product of time resolution and velocity. Relative to frequency and velocity, the size detection resolution is larger, making it more robust against noise but also prone to larger detection errors. Furthermore, experimental results reveal relationships between droplet parameters and input parameters. For example, when fixing the water phase (CH2) and adjusting the oil phase (CH1), the droplet velocity gradually decreases

with increasing input pressure, while the frequency exhibits a trend of initially increasing and then decreasing. The droplet size, on the other hand, gradually increases when fixing the oil phase (CH1) and adjusting the water phase (CH2). Additionally, representing the relationship between input parameters and droplet parameters in a two-dimensional heatmap format allows for a visual assessment of the current performance of the droplet microfluidic chip. For further details, please refer to the supplementary materials.

**Table 1: Frequency and velocity error for different methods.**

| Data | CH1 | CH2 | Method | Frequency error | Velocity Error |
|------|-----|-----|--------|-----------------|----------------|
| 1 | 112 | 100 | Spike-based | 2.4925 | 1.3003 |
|   |     |     | Image-based | 2.4615 | 14.0818 |
| 2 | 110 | 102 | Spike-based | 5.3140 | 7.5342 |
|   |     |     | Image-based | 5.2894 | 24.4896 |
| 3 | 112 | 104 | Spike-based | 2.1263 | 5.8997 |
|   |     |     | Image-based | 2.0927 | 17.463707 |
| 4 | 114 | 106 | Spike-based | 5.4879 | 4.7059 |
|   |     |     | Image-based | 5.4059 | 8.4376 |
| 5 | 112 | 100 | Spike-based | 2.4925 | 1.8696 |
|   |     |     | Image-based | 2.4615 | 4.0531 |
| 6 | 122 | 110 | Spike-based | 1.7795 | 8.2184 |
|   |     |     | Image-based | 1.8441 | 17.8120 |
| 7 | 120 | 112 | Spike-based | 0.5143 | 2.2569 |
|   |     |     | Image-based | 8.1680 | 11.2864 |
| 8 | 128 | 114 | Spike-based | 7.9781 | 6.2824 |
|   |     |     | Image-based | 8.0475 | 33.5221 |
| 9 | 126 | 116 | Spike-based | 8.7242 | 3.3388 |
|   |     |     | Image-based | 8.6589 | 7.0166 |
| 10 | 128 | 118 | Spike-based | 5.0456 | 6.0700 |
|    |     |     | Image-based | 4.8788 | 56.1947 |

## 5.4 Comparsion with image-based method

To further validate the performance of our method, we compared the accuracy and efficiency of the direct spike-based droplet parameter estimation method with the image-based droplet parameter estimation method using real-world data. The bandwidth of actual data transmission is typically limited. Different transmission bandwidths imply different acquisition speeds. To achieve real-time droplet parameter estimation, it's crucial to minimize the required bandwidth for data transmission. Therefore, in this experiment, we set the bandwidth to the maximum required for spike camera transmission and compared the performance of the spike-based parameter estimation method with that of the image-based method under the same bandwidth. In this scenario, the spike-based acquisition speed can reach 20,000fps, while the image-based acquisition speed is only 2500fps. We reconstructed the spike stream into images with a sampling rate of 2500fps to simulate the actual image acquisition results. Then, we transferred the spike-based droplet parameter estimation method to the image domain to perform image-based droplet parameter estimation and compared the results with those directly obtained from the spike. The results, as shown in Table 1, indicate that both the image-based method and the spike-based method exhibit consistent performance in droplet frequency estimation, with small errors compared to actual observations. However, in droplet velocity estimation, the error of the image-based parameter estimation method is significantly greater than that of the spike-based method. This is mainly due to two reasons: On the one hand, limited transmission bandwidth restricts the temporal resolution of image acquisition, leading to decreased accuracy in velocity representation. On the other hand, exposure time makes motion blur more likely in high-speed droplet scenes, affecting the accuracy of velocity estimation.

This result demonstrates that the spike-based droplet parameter estimation method can achieve better parameter estimation with lower bandwidth requirements. In addition, we tested the time consumption of our method for a single evalution, which is approximately 20ms without acceleration. Subsequently, with accelerated optimization, further improvements are achievable.

## 6 CONCLUSION AND DISCUSSION

In this paper, we introduce spike camera into droplet microfluidics for the first time and propose a real-time droplet parameter estimation method based on spike stream, referred to as M-RTDE. By leveraging the advantages of spike camera in achieving full-time scene information recording with lower bandwidth, our method enables real-time parameter estimation in droplet microfluidics and can also recover droplet image information at any given moment. This innovation breaks the traditional separation of acquisition and analysis, as well as the offline control paradigm. Our approach significantly enhances the convenience of droplet microfluidics applications and opens up new directions for related research and applications. To validate the performance of this new method, we collected a dataset called M-SDED, which contains simulated and actual data with different droplet sizes, frequencies, and velocities. Experimental results based on M-SDED demonstrate the outstanding efficiency and accuracy of our method.

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
