# OpenReview forum: "Real-time parameter evaluation of high-speed microfluidic droplets using continuous spike streams"
_acmmm.org/ACMMM/2024/Conference — MM2024 Poster_

### Official Review · Reviewer_F59F · 2024-05-18

**Rating:** 4
**Confidence:** 1

**Summary:**

This paper proposes real-time droplet parameter evaluation using spike camera for microfluidics and Introduces M-RTDE method for real-time droplet parameter estimation.

**Strengths:**

The author claims that (if true) it is the  first to introduce spike cameras to information acquisition in droplet microfluidics.
The proposed spike-based droplet parameter estimation method is interesting.
The spike-based droplet datasets are a significant contribution to the community.
The improvements of experimental results compared to other methods are superior.

**Limitations:**

The paper lacks an introduction to the comparative methods, leading to reader confusion. It's unclear what "Image-based" refers to and whether there are additional methods for comparison. The experiments are also incomplete, needing more datasets and lacking ablation studies.

Can the proposed method balance the advantages of previous imaged-based methods?

**Suitability:**

2

---

### Official Review · Reviewer_CfzK · 2024-05-23

**Rating:** 5
**Confidence:** 2

**Summary:**

The paper introduces a system for real-time parameter estimation of droplet microfluidic control systems in high-speed scenarios. The key innovation of the proposed system is the use of spike cameras, which provide high temporal resolution and dynamic range with a relatively low data rate. To process the spike stream, the authors introduce methods for evaluating droplet generation frequency, velocity, and size. These methods are evaluated using both synthetic and real-world data sets. The results show that the proposed spike-based system parameter estimation is consistent with theoretical expectations and outperforms image-based methods while requiring much less data.

**Strengths:**

- The paper introduces the first spike-based system for real-time parameter estimation of droplet microfluidic control systems. It provides accurate, real-time parameter estimating while requiring significantly less data than state-of-the-art solution.
- The proposed system is evaluated using both synthetic and real-world datasets.

**Limitations:**

-The methods in Section 4 are introduced without clarifying whether they address novel research challenges or are simply a straightforward application of known results.

**Suitability:**

3

---

### Official Review · Reviewer_eXxV · 2024-05-26

**Rating:** 5
**Confidence:** 2

**Summary:**

Microfluidic droplet devices have broad applications due to their high throughput and low power consumption. However, the lack of real-time methods for accurately estimating droplet generation parameters has largely kept droplet microfluidic systems under offline control, making effective feedback during droplet generation a challenge. In this paper, the authors integrate a spike camera into the droplet collection system, efficiently capturing information with a spike stream. Based on this, the authors propose a spike-based real-time droplet generation parameter estimation algorithm, achieving real-time estimation of microfluidic droplet parameters.

**Strengths:**

1. The aim of this paper is to achieve real-time and relatively high-precision parameter estimation for droplets under relatively low bandwidth conditions. Spike cameras, known for their high dynamic resolution, provide an intuitive and effective method for integrating into droplet microfluidic information collection. The experimental results confirm this.
2. For droplet microfluidic information processing under a spike camera, the authors propose an end-to-end spike-based droplet parameter estimation method, which has demonstrated good performance in experiments.
3. The authors have collected and open-sourced the first spike-based droplet dataset (SDD), which I believe contributes to the research community in this field.

**Limitations:**

I see no major flaws in this paper and I am inclined to accept it.

I believe that the method is valid, nevertheless, I am not sure about the extent to which it contributes to the field as a whole, i.e. how widely it is applied and how much it contributes to the field as a whole. I therefore need a reviewer who is more experienced in this area to help me make a judgement. My rating at this stage is ‘Weak Accept’.

**Suitability:**

2

---

### Official Review · Reviewer_eCjb · 2024-05-26

**Rating:** 4
**Confidence:** 3

**Summary:**

The paper presents a novel method for real-time parameter evaluation of high-speed microfluidic droplets using continuous spike streams. The authors introduce spike cameras into droplet microfluidics for the first time, proposing a method named RTDE that integrates spike cameras into the droplet collection system. This method allows for efficient capture of information via spike streams and employs a spike-based estimation algorithm for real-time droplet generation parameters. The performance of RTDE was validated using both synthetic and real-world datasets, demonstrating high precision and superior time efficiency compared to image-based parameter estimation methods. The spike camera's ability to offer high temporal resolution with low data throughput makes it a promising tool for real-time, precise parameter estimation in droplet microfluidics.

**Strengths:**

1)The paper introduces a pioneering use of spike cameras in microfluidics, showcasing a new direction for real-time parameter estimation.
High Precision: Experiments on both synthetic and real-world data indicate that the proposed method achieves parameter estimations that closely match ground truth values.
2)The spike-based method demonstrates exceptional temporal efficiency, capable of meeting real-time processing requirements.
3)The spike camera's design allows for high temporal resolution with relatively low bandwidth, which is advantageous for real-time applications.

**Limitations:**

1) Insufficient Justification for Event Cameras' Limitations: The paper argues that spike cameras are preferred over event cameras for high-speed microfluidic droplet parameter estimation due to their ability to capture high-quality images at any given moment alongside real-time parameter estimation. However, the justification provided for event cameras' supposed inadequacies in this context is not sufficiently robust.

2) Typographical Errors: The manuscript contains typographical errors within the mathematical expressions of Section 4.2, particularly impacting Equations 9 and 13. These mistakes misguide readers who are trying to comprehend the underlying mathematical logic. Incorrect Term Usage in Equation 9: In Equation 9, there is an error where the term \(\hat{y}_1\left(t\right)\) has been mistakenly written as \(\hat{y}_2\left(t\right)\). Equation 13 features a misuse of parentheses that could cause a misinterpretation of the formula.

3) Lack of Consistent Acronym Reference: The manuscript introduces the acronyms M-RTDE and M-SDED only in the conclusion section, with no prior mentions throughout the rest of the paper. This inconsistency in terminology can lead to confusion for readers who may not immediately connect these acronyms to the methods and datasets discussed earlier in the document.

**Suitability:**

2

---

### Meta-Review · Area_Chair_h7P8 · 2024-07-01

**Recommendation:** Accept (Poster)
**Confidence:** 5

**Metareview:**

This paper introduces RTDE, a real-time evaluation method for high-speed droplet parameters using a spike camera integrated into a droplet collection system. Leveraging the neuromorphic capabilities of the spike camera, the proposed method captures high temporal resolution scenes with low data throughput. The authors develop a spike-based estimation algorithm to process the spike stream and estimate droplet generation parameters in real-time. Validation of RTDE is conducted on a spike-based droplet dataset (SDD) consisting of synthetic and real data with varying flow velocities, frequencies, and droplet sizes. Experimental results demonstrate that RTDE achieves high precision in parameter estimation and significantly outperforms image-based methods in terms of time efficiency. The RTDE method presents a significant advancement in the field of microfluidics by introducing a novel, efficient, and accurate approach to real-time droplet parameter estimation. The innovative use of spike cameras, coupled with a well-designed spike-based estimation algorithm, addresses a critical challenge in droplet microfluidic systems. The experimental validation is comprehensive, demonstrating the robustness and practical advantages of the method. Despite minor areas for improvement in algorithmic clarity and hardware generalizability, the paper's contributions are substantial and well-aligned with the interests of the ACM MM community. All reviewers give positive reviews, this paper could be accepted.